# Significantly Improved COVID-19 Outcomes in Countries with Higher BCG Vaccination Coverage: A Multivariable Analysis

**DOI:** 10.3390/vaccines8030378

**Published:** 2020-07-11

**Authors:** Danielle Klinger, Ido Blass, Nadav Rappoport, Michal Linial

**Affiliations:** 1Department of Biological Chemistry, Institute of Life Sciences, The Hebrew University of Jerusalem, Jerusalem 91904, Israel; danielle.klinger@mail.huji.ac.il; 2The Rachel and Selim Benin School of Computer Science and Engineering, The Hebrew University of Jerusalem, Jerusalem 91904, Israel; ido.blass@mail.huji.ac.il; 3Department of Software and Information Systems Engineering, Faculty of Engineering Sciences, Ben Gurion University of the Negev, Be’er Sheva 84105, Israel

**Keywords:** epidemiology, SARS-CoV-2, multivariable regression, tuberculosis, demography, coronavirus, MMR vaccine

## Abstract

The COVID-19 pandemic that started in China has spread within 3 months to the entire globe. We tested the hypothesis that the vaccination against tuberculosis by Bacille Calmette–Guérin vaccine (BCG) correlates with a better outcome for COVID-19 patients. Our analysis covers 55 countries complying with predetermined thresholds on the population size and number of deaths per million (DPM). We found a strong negative correlation between the years of BCG administration and the DPM along with the progress of the pandemic, corroborated by permutation tests. The results from multivariable regression tests with 23 economic, demographic, health-related, and pandemic restriction-related quantitative properties, substantiate the dominant contribution of BCG years to the COVID-19 outcomes. The analysis of countries according to an age-group partition reveals that the strongest correlation is attributed to the coverage in BCG vaccination of the young population (0–24 years). Furthermore, a strong correlation and statistical significance are associated with the degree of BCG coverage for the most recent 15 years, but no association was observed in these years for other broadly used vaccination protocols for measles and rubella. We propose that BCG immunization coverage, especially among the most recently vaccinated population, contribute to attenuation of the spread and severity of the COVID-19 pandemic.

## 1. Introduction

COVID-19 has spread within 3 months to 213 countries across the globe. The country-specific reports that are compiled daily by the World Health Organization (WHO) and made publicly available, provide statistical information on the number of tests performed, the number of confirmed cases, deaths and the cumulative state of patients hospitalized in serious and critical conditions [1]. Along with the spread of the pandemic, most countries imposed a policy of social distancing and other regulation to mitigate COVID-19 [2,3]. Despite the intense effort, key epidemiological parameters are still missing [4,5,6,7,8,9]. With 400,000 reported deaths and a world average of 51 deaths per million (DPM, June 6th, 2020), the death toll remains the most reliable measure for monitoring the spread and progression of the disease across countries [10]. While some European countries such as Belgium and the UK the DPM is >500, other infected countries (e.g., Hungary, Norway) are closer to the world average. The large differences in COVID-19 outcomes, even among neighboring countries (e.g., Spain and Portugal), are not likely to solely reflect differences in the regulations imposed by each country at the initial phase of the pandemic [11,12].

In this study, we tested the possibility that the extent and spreading of COVID-19 cases are associated with the status of tuberculosis (TB) immunization across the world. The Bacille Calmette–Guérin vaccine (BCG) contains a live attenuated strain of *Mycobacterium bovis*, is widely used to eradicate TB and was among the most broadly used vaccinations in the 20th century in neonatal and young children [13,14]. Currently, the BCG vaccine is provided to the entire population in most countries with high TB incidence [15]. Over the last two decades, numerous countries changed their policy and restricted BCG immunization policy to non-native born migrants from high TB burden countries [16]. Notably, numerous epidemiological and immunological studies demonstrate that BCG vaccination results in reduced morbidity and mortality to subsequent infections, presumably by its effect on the immune response [17]. Specifically, we questioned whether BCG vaccination regimens in different countries are linked with different COVID-19 outcomes. Our analysis considered a broad range of variables covering demography, economy, medical status, health system strength, and dynamics of the lockdown policy. We found that the BCG admission coverage of the young population is inversely correlated with COVID-19 outcomes. The implication of these observations on national policy for immunization is discussed.

## 2. Materials and Methods

### 2.1. Data Extraction

Information regarding COVID-19 outcomes was extracted daily between 29 January and 21 May 2020 from the Worldometers website [18] using a crawler written in Python software version 3.7 (https://www.python.org/downloads/release/python-370) [19].

Demographic measures of countries were extracted from the Worldometers website on 17 April 2020 [20]. Information regarding the share of population >65 years and economic development indicators were extracted from the World Bank data [21]. Information on educational management and school closure, as a measure of the quarantine status of the country, was extracted from the UNESCO institute of statistics dataset [22]. Prevalence of chronic diseases (e.g., obesity, type 2 diabetes) and the death rate from cardiovascular disorders were extracted from Our World In Data (ourworldindata.org) website. Appendix A provides the source for this country-related information.

Information regarding past and present BCG administration practices in every country was extracted from the BCG world atlas [20]. Two vaccination status groups were considered: (i) countries that had either a current or past national mandatory vaccination policy (49 in total), (ii) countries that have only administered BCG vaccinations to specific groups at risk (6 in total). In the latter group, only a negligible fraction of the population is BCG vaccinated [20]. In addition, the estimates for BCG, measles and rubella vaccination coverage between years 1980 and 2018 were extracted from the annual WHO reports [1]. For additional resources used to establish the years of mandatory BCG administration see Appendix A.

### 2.2. Data Analysis and Statistical Tests

Countries were normalized by accounting for their population size (per 1 M, PM). The normalized COVID-19 outcomes that were considered are death (DPM), positively validated cases (cases per million (CPM)), hospitalization with serious and critical conditions (SPM) and recovered (RPM). Accounting for the varying stages of the pandemic in each country, we define a unified aligned key date of a country as the first date when DPM reached for the first time the DPM value of 0.5 or higher. The following analysis was conducted across changing dates following the key aligned date (at a range of 10–50 days). In binary or categorized tests, we applied the ranked Wilcoxon test. For the continuous data, we applied linear regression. The regression fit and the calculated statistical significance (*p*-value) for the COVID-19 outcomes are reported. We tested the correlations between outcomes and years of BCG administration using Pearson’s correlation, and reported the analytic *p*-values as well as permutation tests’ *p*-values. Correlation robustness test was performed using repeating sub-sampling 2000 times 90% of countries. We report the high fraction of tests where correlations were significant (*p*-value < 0.05). A correlation between the BCG by age groups was determined by partitioning the population of each country into three groups: (i) 24 years and younger; (ii) 25 to 64 years; (iii) 65 years and older. From the age partition and the BCG coverage within each age group, a value that measures the percentage of the population weighted by the share of the age group with BCG is calculated. All analyses were conducted in R software version 3.5.2 (https://cran.r-project.org/bin/windows/base/old/3.5.2/) [23].

Further details on the statistical approach and the data processing are available in Text S1.

## 3. Results

### 3.1. BCG Administration Years are Negatively Correlated with COVID-19 Outcomes

In order to increase the robustness of the analysis, countries were included in the selected cohort if their population size was >3M, and they met the criteria of ≥3 deaths per 1 M population on 17 April 2020. Altogether, there are 134 countries with population size >3M. Among them, 55 complied with both thresholds, covering 62.9% of the world population. A regional partition of these countries is shown in Figure 1. For detailed information on the countries included in the analyses, see Appendix A.

First, we analyze COVID-19 outcomes as the difference in deaths or cases per million (DPM and CPM, respectively). Thus, the analysis was performed 20 days following two different alignments of key dates (defined by DPM ≥ 0.5 and DPM ≥ 2). Figure 2 shows strong and significant correlations between COVID-19 outcomes and the number of years of BCG administration. We observed a strong negative correlation with DPM outcome with R = −0.48 (*p*-value = 0.00056) and −0.47 (*p*-value = 0.00084) when aligned at DPM threshold of 0.5 and 2, respectively (Figure 2a,b). Similarly, for the CPM as COVID-19 outcome, we observed a similar trend with R = −0.38 (*p*-value = 0.0091) and −0.35 (*p*-value = 0.017) when aligned at DPM of ≥0.5 and DPM ≥ 2, respectively (Figure 2c,d).

To test the generality of our observations we repeated the analysis at a broad range of time points along with the progress of the disease, starting from the 10th-day post alignment and showing the trends in 10-day intervals (10 to 50 days, Figure 3). For this analysis, we tested the outcomes of COVID-19 confirmed serious/critical cases (SPM) and the number of recovered (RPM), in addition to the DPM and CPM. The results of the DPM and SPM show a highly significant association for all time points, corroborated by the robust results obtained from performing 2000 permutation tests for each time interval for all tested outcomes (Appendix A). An additional test for robustness was performed by repeating the 2000 correlation tests on 90% random subsamples of countries. We found that 4/5 of the DPM- and SPM-examined dates presented significant results in more than 95% of random tests (Appendix A).

### 3.2. Multivariable Analysis Reveals a Strong Contribution of BCG Administration to the COVID-19 Outcome Statistics

Countries differ in many quantitative measurements like population size, Gross Domestic Product (GDP), lifespan, median age and more. To control for some of the potential confounding factors, we included numerous demographic values for a multivariable linear regression. The analysis included 23 demographic, economic, pandemic restriction-related and health-related country-based variables. The results show that the number of BCG administration years ranks consistently within the top two most significant coefficients and is within the top coefficients with the larger effect (as measured by the normalized beta coefficient, out of 23 coefficients) (Figure 4). The results are consistent among the different times observed (for further analysis see Appendix A). Notably, a strong positive beta coefficient value is associated with the median age. This may be due to the fact that countries with a higher median age, a parameter reflective of the lifespan and the demographic bulges [2], have a higher fraction of older residents. Since a substantially higher risk of death is associated with infected older populations, countries with a higher median age are naturally susceptible to a higher percentage of deaths [24,25]. In accordance with COVID-19 age-associated risk, in the multivariable analysis the median age was associated with a strong positive coefficient. In addition, cancer percentage is also significant, and may reflect a confounding factor for lifespan and the rarity of cancer occurrence in the young population. The combined contribution of gender, chronic disease prevalence, and economy to the spread and fatality of COVID-19 was already reported [2], and will not be further discussed.

### 3.3. Highest Correlation with BCG Age Coverage Applies to the Most Recently Vaccinated

Due to the varying effect of each age group on the viral spread in the population, we next investigated the relevance of age groups to the observation showing that years of BCG administration are strongly correlated with better COVID-19 outcomes. Epidemiological studies from COVID-19-positive cases in Shenzhen China confirm the importance of the young group age in the spread of the disease [26]. Specifically, children (1–16 years) were considered fundamental in the chain of transmission [27]. Figure 5 shows the correlation of total years of BCG administration with DPM difference according to the country-based age composition. The population in each country was partitioned to young (<24 years of age), working-class (25–64 years) and old (>65 years). The correlation with the young age group (tested at 20 days post-alignment key dates) shows the highest significance with R = −0.54, *p*-value = 7.6 × 10^−5^ (Figure 5a), and the correlation with the age group of 25–64 years (at 20 days) is also significant with R = −0.32, with a weaker significance (*p*-value = 0.028, Figure 5b). Both correlations remain significant throughout a 50-day period post alignment. Remarkably, for the old age group (>65 years), at all the time-frames tested, the correlation was negligible and insignificant (Figure 5c).

**Notably, the age composition varies across countries.** To examine the robustness of the results, we performed the same correlation analysis while not accounting for the actual fraction occupied by each of the age groups. The results (20 days post alignment date) are very similar to those obtained by weighting the fraction of the different age groups. Specifically, the correlation for the young age group is R = −0.58, *p*-value = 2 × 10^−5^; middle age group is R = −0.35, *p*-value = 0.016 and the old age group is R = −0.14, *p*-value = 0.34. We further tested the statistical significance for the other outcomes (SPM, CPM and RPM), by age group according to population share, along different time points. Figure 5d–f show the dominant contribution of the young age group to the negative correlation at 10–50 days post-alignment. Notably, the outcome of recovered per million (RPM) has a significant negative correlation only among the young group. The drop in RPM significance from 30 to 50 days is consistent with the epidemiological survey reporting on the long-time gap until recovery [28]. The middle age group (Figure 5b,e) is mostly insignificant and shows a borderline significance for the DPM and SPM as outcomes. All observations regarding the elderly (Figure 5c,f) are insignificant. We conclude that the elderly group does not contribute to the strong correlation with BCG administration.

The pronounced signal in the young age group led us to investigate whether recent immunization may have a positive effect on the outcome. We divided the countries into three disjoined groups representing their vaccination policies over the past 15 years, disregarding the population share of the 0–15 age group in each country: (i) countries with mandatory immunization policies over the past 15 years; (ii) countries with mandatory immunization policies, which were applied for less than 7.5 years within the past 15 years; (iii) countries with no mandatory immunization policies over the past 15 years (Figure 6a). Applying a test with DPM outcome, yielded highly significant results across all tested post-alignment (10–50 days) dates, establishing that countries with BCG immunization policies over the past 15 years have a significantly lower rate of DPM with respect with countries in group (iii).

The significant result in the young age group raised the question of whether other immunizations might have a significant effect. To this end, we tested COVID-19 outcome and the globally used immunization against measles and rubella. We divided the countries into two groups representing their vaccination coverage over the past 15 years (as provided by the WHO): (i) countries with above median coverage; (ii) countries with below median coverage. While the world overall coverage of MCV1 (measles containing vaccine) is high in recent years, with an average of 88% in 2017, as provided by the WHO, the MCV1 coverage from 1990 shows substantial variation (e.g., 73.8% in India, 80.2% in Italy, 85.7% in Algeria, 89.2% in Belgium) [1]. Applying the Wilcoxon test, yielded insignificant results at all tested dates (10–50 days post alignment). Opposite to the BCG results, we found no correlation between the degree of Measles and Rubella vaccination coverage and COVID-19 outcomes (Figure 6b,c).

### 3.4. Data and Materials Availability

All data needed to evaluate the conclusions in the paper are present in the paper and in the Supplemental Materials. An online tool for displaying the analytical results is available at: https://covi.shinyapps.io/COVID19/. It is a useful analytical webtool for single variant statistics, correlations, multivariable analyses and more. The user-friendly platform allows the changing of parameters by setting a threshold on population size, the time along the pandemic progression, selecting predetermined outcomes as a reference date for the alignment and changing the thresholds for alignment date. The code and data are available at: https://github.com/nadavrap/COVID19. Additional data and support related to this study may be requested from the authors.

## 4. Discussion

The significantly strong correlation between the BCG vaccination and better outcomes for COVID-19 is shown across many countries, covering the majority of the world population (Figure 1 and Appendix A). The findings are based on an unbiased view of all countries that comply with predetermined thresholds for DPM and population size (see Materials and Methods).

The strong negative correlation between the BCG administration years and DPM was sustained at a range of time-points from the aligned-date (Figure 3). For testing the stability of the DPM correlation, we repeated the analysis for additional COVID-19 outcomes. We consider the number of hospitalized people (at a specific date) which were indicated by a serious or critical condition (SPM). This measurement is strongly dependent on the health care capacity and the actual phase of the pandemic. Using the SPM rather than the DPM as a measure shows that the trend of the BCG administration year remains stable and significant (Figure 3). As expected, the negative correlation to COVID-19 validated positive cases (CPM) is weaker relative to DPM. CPM is likely to reflect the capacity of different countries to carry out reliable molecular tests (PCR-based) or clinical tests (lung CT pathology) [29], and the national policy for targeted testing [30]. We observed no significant correlation for the country-level number of recovered (RPM). We attribute it to the non-standardized definition for COVID-19 recovery [25], the time delay for confirmed recovery [31]. Altogether, RPM is the least reliable outcome as the pandemic peak ranges greatly among countries.

Our multivariable analysis highlights the strength of combining a broad range of country-based quantitative observations. Among the analyzed variables are the economic measures [32], health system capacity (e.g., doctors per 1000 people), population composition, exposure to infectious diseases (e.g., prevalence of TB), pandemic restriction-related measures (i.e., school closure dates), major comorbidities (e.g., cancer, diabetes) and habits (e.g., smoking by sex). Many of these measures are correlated and may reflect confounding factors. Most importantly, the multivariable analysis validated the importance and statistical significance of BCG immunization years given all other variables (Figure 4). Policy toward quarantine, enforcement of isolation regulation (e.g., closure of cultural events, public transportation) were implemented at a country-based time point. For example, we included the number of days of closing the educational facilities relative to country alignment date as a variable in the analysis. It was used as a proxy for the level of the constraints imposed along the pandemic progression [33]. While it is expected to have a strong impact on COVID-19 spread [25,34], this variable did not contribute to rejecting the hypothesis, and had a minimal impact on the multivariable analysis (Figure 4).

The exact date of BCG administration within each country, combined with the actual immunization coverage (provided by the WHO) and the population structure, allowed us to explicitly test the effect over time of the BCG immunization. Specifically, partitioning the population to young group, middle-age group and elderly confirmed that the strongest signal towards COVID-19 outcome is associated with young (<24 years, Figure 5) and slightly also to the middle age group (25–64 years). However, the elderly (>65 years) that are at the highest risk for COVID-19 mortality do not correlate with BCG higher coverage. The implication of our observation for COVID-19 epidemiology and pandemic dynamics is evident [25].

Our results suggest that in countries where the young population is vaccinated by BCG, a maximal protection is provided to the whole population. In countries where the young and middle-aged population groups were vaccinated, a significantly lower number of cases and deaths in the total population were observed (Figure 5). Universally, the DPM among young people is very low (0.04% for <17 years) [18]. Therefore, the main contribution of the young age is with regards to the impact on the chain of infection [27,35,36]. Children and young adults (tested for ages 0–22 years) tend to make more social contacts than adults and hence are likely to contribute more to transmission than adults [35,37]. The middle-age group (25–64 years) overlaps with the group that is specified by an extensive cross-generation social interaction [37,38]. Thus, the higher BCG coverage of the young and middle-aged groups is associated with the attenuation of overall infection rate. The lack of correlation between BCG coverage for the elderly (>65 years) and COVID-19 outcomes is in accord with the negligible impact of this population on viral transmission to the community [27,35].

Several reports proposed that BCG-vaccinated populations are resistant to viruses, and in particular toward SARS-CoV-2 [39]. Despite the broad usage of BCG for almost a century, and the underlying mode of action, the indirect long-term effect of BCG on the immune system remains enigmatic [40,41,42]. It is postulated that the positive effect of BCG immunization on COVID-19 outcomes is achieved by an improved systemic immunity which applies to the most recently vaccinated group. Accordingly, the younger age group that was recently vaccinated is likely to benefit from an immunological protection whereas the older age group (that was vaccinated by BCG over 65 years ago) are unlikely to display BCG-driven immune protection. We found no correlation between the degree of measles and rubella vaccination coverage and COVID-19 outcomes (Figure 6). These immunizations were shown to have some reduced susceptibility to viral infection [43].

There are several limitations that need to be addressed in supporting our main findings and the conclusions from this study. The first difficulty stems from the fact that countries vary greatly by their area, population density and age structure that can mask the apparent BCG protective effect [44]. Moreover, difference in culture and habits (e.g., religious gathering, social distancing, smoking), economy, demography and the capacity of the health system often cannot be easily generalized. In addition, our results may potentially be driven by a small number of influential countries.

To address some of these inherent difficulties, we duplicated the analyses for countries that were bounded by population size (>3 M and <100 M). We observed no effect on the main findings as shown in Appendix A. Moreover, an analysis that relies on COVID-19 static information is likely to suffer from unstable findings due to unpredicting trends on the pandemic dynamic [45,46]. By altering the threshold for the alignment date, we confirmed the robustness of the results (Figure 2). In addition, we tested for significance using sub-sampling tests. When randomly sampling 90% of the countries (with 2000 repeated tests), we found the 4/5 of the DPM- and SPM-examined dates presented significant results in more than 95% of random tests (Appendix A).

The underlying mechanism by which BCG exerts its beneficial effect on infectious diseases is not fully resolved [47,48]. Recent evidence highlights the importance of reprograming of the innate system [49] and enhancing the response of heterologous T helper cells [50]. Overall, the efficacy of BCG against TB is expected to cover approximately 15 years [51]. Thus, the strong statistical significance value for BCG being most effective for a recently immunized population argues that the active immunization phase rather than a residual non-specific protection from early-life events is associated with a better COVID-19 outcome [52,53].

Knowledge on the long-lasting effects of BCG on the immune system is essential for designing in-vivo experiments and ultimately effective vaccines. Some of the reported inconsistency with respect to the BCG-induced immune response was attributed to differences in the BCG strains, manufacturing methodology, and route of administration [54]. Several in-vitro assays showed that different sources of BCG are associated with a range of clinical efficacies [55]. Moreover, an improved protection for TB in rhesus macaques was reported for BCG that was admitted by repeated pulmonary mucosal delivery rather than by the default injection protocol [56]. We propose that future clinical trials for testing the impact of BCG on COVID-19 should consider the strain origin and the modes of BCG immunization.

Our results cannot exclude the possibility that a “pre-trained” state of immunity by BCG immunization exerts its positive effect, thus improving COVID-19 outcome at a population level. The finding that shows a strong and robust association of BCG coverage in the young age group with improved COVID-19 outcomes in the whole population calls for ongoing monitoring of the evolution of the pandemic world-wide. To this end, we developed a user-friendly platform with the capacity to change any of the dynamic parameters according to the pandemic progression.

## 5. Conclusions

We conclude that the inverse correlation with BCG administration years, the impact of a recent vaccination, and the validated role of the young population in the spread of COVID-19 calls for revisiting the global and national BCG immunization policy [27,35,37]. A growing number of clinical trials for testing the efficacy of BCG vaccination have been initiated. In one such trial, BCG was admitted once a month for 3 consecutive months in the elderly (60–75 years old) to test the prevention of acute upper respiratory tract infection. The results show significant prevention of infection in parallel to an improved response of T-helper cells [57]. Several on-going randomized controlled clinical trials cover different populations and several BCG strains. In one trial (named BADAS, USA), 1800 participates will be introduced to the BCG Tice strain, while in a larger trial (called BRACE, Australia) there will be over 10k healthcare workers tested by the Danish strain 1331. The goals in all these clinical trials are to determine if BCG vaccination reduces the incidence and severity of the COVID-19 pandemic [58,59]. While the WHO does not recommend BCG vaccination for prevention of COVID-19, we anticipate that the results of the BCG immunization clinical trials will provide guidelines for better controlling COVID-19 spread and severity. Without derogating from the importance of the results of clinical trials on the individual level, our opinion is that they will not necessarily directly affect country-wise epidemiological decisions. The reason is that minor results in clinical trials (such as 5%) do not have the same effect at the individual level, in comparison with the population level—where even minor reductions in viral transmission can have major impact in terms of the pandemic’s spread, population morbidity and mortality toll.

## Figures and Tables

**Figure 1 vaccines-08-00378-f001:**
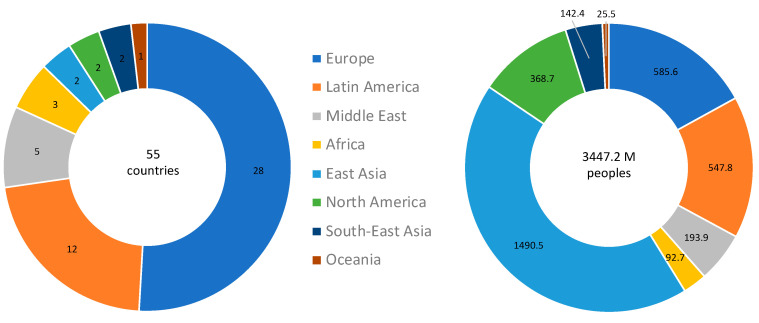
Countries analyzed in this study by geographical regions. Countries that comply with the predetermined thresholds for the population size and a minimum rate of deaths per million (DPM) at the analysis date are included. These countries (55 total) are partitioned by their geographical regions (**left**) and the cumulative population within each region (**right**).

**Figure 2 vaccines-08-00378-f002:**
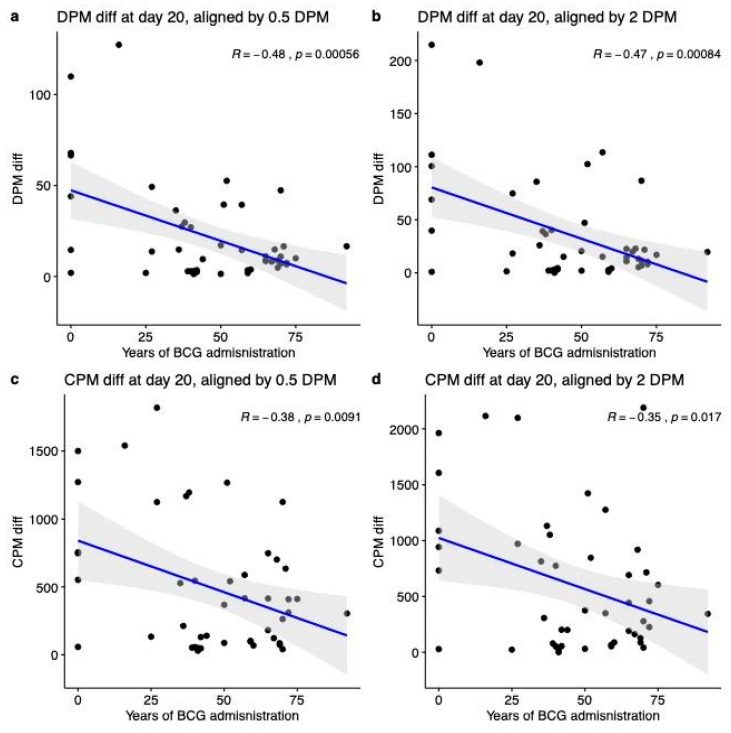
Statistical analysis of COVID-19 outcomes and years of Bacille Calmette–Guérin vaccine (BCG) administration. All correlations were measured at 20 days following the alignment key date. Correlations of years of BCG administration with DPM = 0.5 (**a**) and DPM = 2 (**b**). Correlation with cases per million (CPM) diff. at 20 days when the key date was defined as CPM = 0.5 (**c**) and CPM = 2 (**d**). DPM diff. and CPM diff. are calculated by the differences in the numbers from the measured date to alignment date. Shaded areas represent the 95% confidence intervals.

**Figure 3 vaccines-08-00378-f003:**
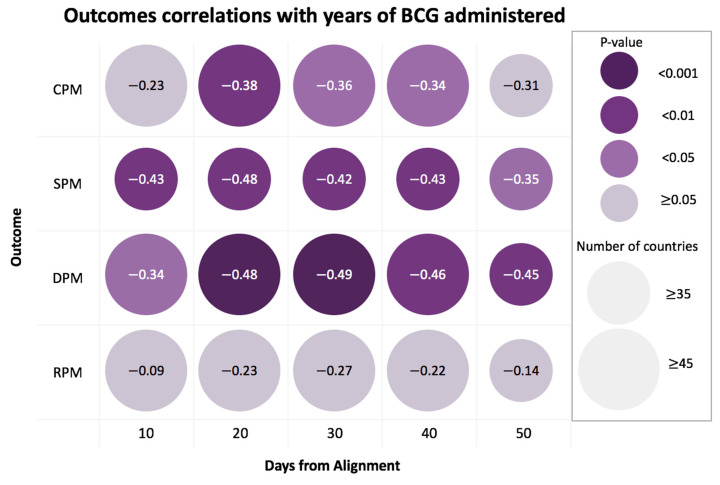
COVID-19 outcomes’ correlations with years of BCG administration. Each row represents a different outcome. Each column represents the time interval of the outcome from the alignment date. Circle size depicts the number of countries, and color represents the statistical significance of the correlation with darker purple color indicating a higher significance. Values in circles are the correlation estimates where all the correlations have negative values. Note that while all countries reached 20 days post alignment of DPM ≥ 0.5, at 50 days post alignment, some countries (5) that were still at an earlier phase of the pandemic at the alignment date failed to provide information. DPM, CPM, SPM and RPM stand for the number per million for death, validated cases, serious and critical conditions and recovered, respectively. For further details, see Appendix A.

**Figure 4 vaccines-08-00378-f004:**
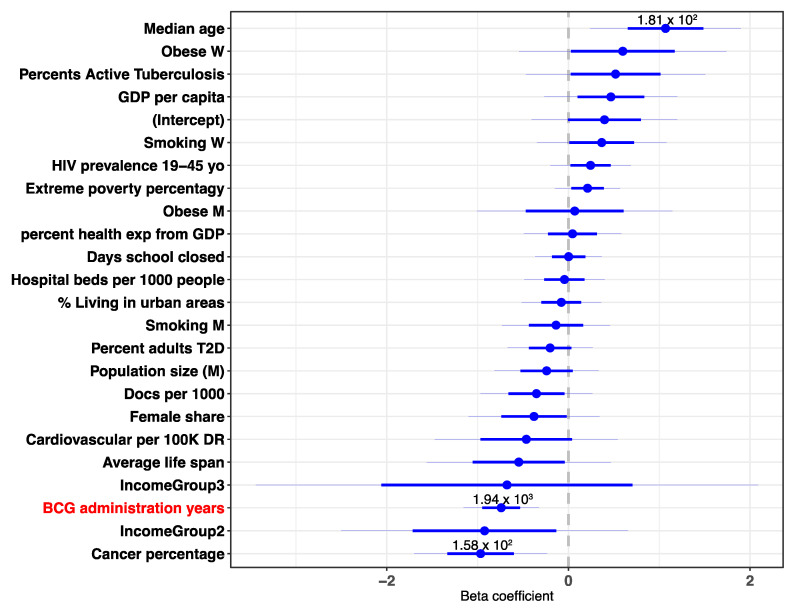
Multivariable analysis of country-centric quantitative data. Beta coefficients of the normalized multivariable linear regression for DPM at day 20 are shown. Blue lines represent the coefficients’ 95% confidence intervals. *p*-values are shown for all variables with *p*-value < 0.1.

**Figure 5 vaccines-08-00378-f005:**
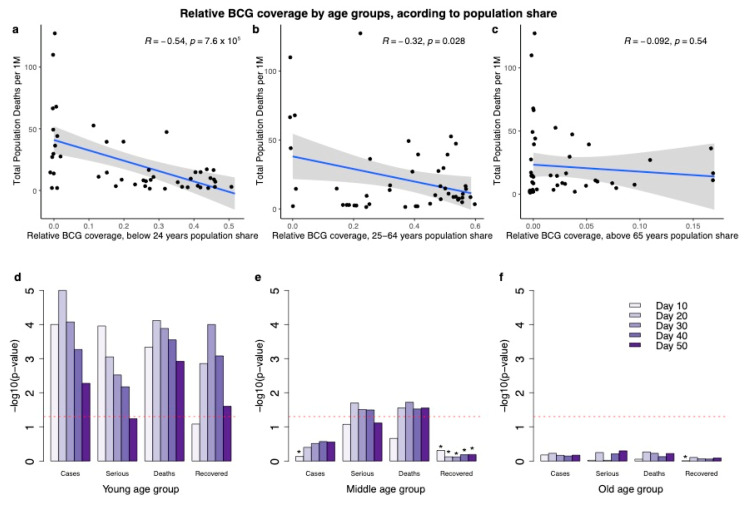
BCG coverage with respect to the DPM difference among three age groups. All correlations (**a**–**c**) and statistical significance (**d**–**f**) were measured following the DPM ≥ 0.5 alignment key date. Relative BCG coverage is partitioned to three age groups, weighted by population share: (a) young (0–24 years), (b) middle age (25–64 years) and (c) old age group (>65 years). Notice that data points in panels a–c were slightly moved horizontally to help distinguish overlapping symbols. The histogram (d–f) shows the statistical significance of the correlation of BCG years of administration for the 4 different COVID-19 outcomes according to the 3 age groups marked as: young (d) middle age (e) and elderly (f). Days from the key alignment date are colored from light to dark purple (10 to 50 days). The statistical significance is shown as −log10 (*p*-value), the dashed red line indicates *p*-value of 0.05. Asterisk represents the outcome with a positive correlation. All results with a positive correlation, marked by asterisks (*), are insignificant.

**Figure 6 vaccines-08-00378-f006:**
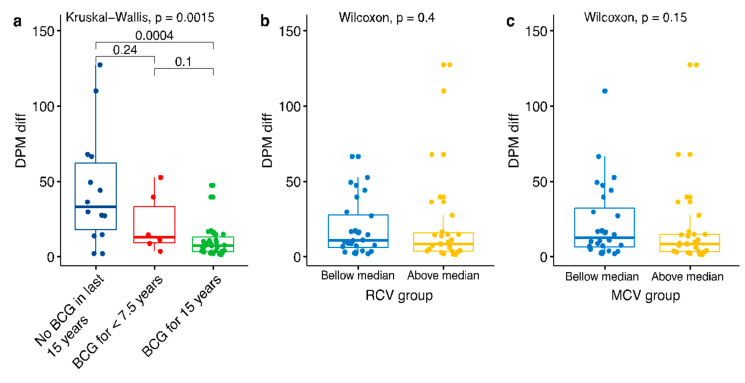
Immunization coverage for the last 15 years. (**a**) The DPM diff. (difference in DPM from the value at alignment key date) in countries that have (green), partially have (red), or have not (blue) rolled immunization BCG programs over the past 15 years. The statistical significance values are shown for each pair. (**b**) The statistics of RCV (rubella) and (**c**) MCV (measles) vaccines and COVID-19 DPM diff. according to the vaccination coverage. The partition of the countries is according to those above (yellow) and below (blue) median % coverage for each of these tested vaccines. Data covering the past 15 years (2004–2018) was extracted from WHO reports. The statistical significance values are listed.

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
