# Peer review of "Significantly Improved COVID-19 Outcomes in Countries with Higher BCG Vaccination Coverage: A Multivariable Analysis"

_vaccines, 2020, doi:10.3390/vaccines8030378_

Round 1

Reviewer 1 Report

The study is well designed and analyzed properly. The manuscript is well written and the results are clearly presented. The discussion and conclusion are appropriate. 

change the "Experimental Section" to "Method".

Author Response

Thanks for the positive comments. We have changed to "Materials and Methods" and edited the manuscript for improving righting and English.  

Reviewer 2 Report

Dear authors.

congratulations for this important study than I am sure will give more important informations about Covid-19 that will help to control the disease at a public health approach. 

I suggest to improve the quality of the figures. Some graphics are too small to be able to be read (eg: line 105 and line 129. Maybe if you put the images bigger they will stay ok. 

I also suggest you to add some information about ethical issues, related with the use of data. Can you put the information about an ethical commitee aprovement?

In the conclusion it would be nice that you could go more far. When you say that "While the WHO does not recommend BCG vaccination for prevention of COVID-19, several clinical trials with BCG are undertaken..." Please let us know more about your opinion regardind the results of your research. You start this point saying that "We conclude that the inverse correlation with BCG administration years, the impact of a recent vaccination, and the validated role of the young population in the spread of COVID-19 calls for
revisiting the global and national BCG immunization policy." But in what way? Reintroduce the BCG as a vaccine in national plans like it was some years ago in some countries? 

Also you can talk a bit about the limitations of your study and the contributions for research, public health, and society. 

You manuscript is already really good. Just a litle more steps to become perfect. I look forward to hear from you.

Author Response

Thank you for the kind words and positive comments. 

I suggest to improve the quality of the figures. Some graphics are too small to be able to be read (eg: line 105 and line 129. Maybe if you put the images bigger they will stay ok.

As suggested we enlaVrged the figures to help the readers to see the details.

I also suggest you to add some information about ethical issues, related with the use of data. Can you put the information about an ethical commitee aprovement?

 All data collected is based on open public domains (this is one of the advantages of this study). Thus, there are no ethical issues involved.

In the conclusion it would be nice that you could go more far. When you say that "While the WHO does not recommend BCG vaccination for prevention of COVID-19, several clinical trials with BCG are undertaken..." Please let us know more about your opinion regardind the results of your research. You start this point saying that "We conclude that the inverse correlation with BCG administration years, the impact of a recent vaccination, and the validated role of the young population in the spread of COVID-19 calls for
revisiting the global and national BCG immunization policy." But in what way? Reintroduce the BCG as a vaccine in national plans like it was some years ago in some countries?

 We added as suggested a more explicit description of "what next" and expanded our view on the relevance of BCG clinical trials.  

Also you can talk a bit about the limitations of your study and the contributions for research, public health, and society.

The limitation of the study was expanded.

We are happy to have  these comments that allow us to speculate and test the role of BCG vaccination on society and human health.

Reviewer 3 Report

Title: Significantly Improved COVID-19 Outcomes in Countries with Higher BCG Vaccination Coverage: A Multivariable Analysis
Authors: Klinger et al

Summary: The authors conduct a statistical analysis evaluating
association of COVID-19 outcomes with years of use of the TB vaccine
BCG across 55 countries. The analysis includes marginal correlation
analyses as well as multivariate regression analyses. The conclusion
is that there appears to be a negative correlation with multiple
disease measures including deaths, especially among the younger
people.

Review summary: The article is well-written and the analyses appear sound.
The authors are careful to communicate their methods well, provide
clear attribution of their data sources, and they
provide a link to an online tool for further analysis of the data
[this reviewer did not review that link]. An additional sensitivity
analysis supports that the results are not driven by inclusion of
small countries. The authors' additional interpretations
about the implications of these results come across to this reader as
overstated, or requiring of additional explanation.

Mostly Major comments:
1) Line 124: please comment on the potential for the result to be
driven by a small number of influential points. It seems from eying
this and other plots that the countries with 0 BCG have higher variation and
higher mean values of the covid outcomes in multiple analyses. Is
there a correlation among the countries with non-zero values? If not,
then perhaps stating it as a general correlation is not sufficiently
clear. If so, it's useful to state this as it is not clear to the
reader from the plots.
2) Line 157-158: This is a subtle point but I think I get it. Median
age reflects both lifespan and demographic bulges, etc. So this
negative coefficient could indicate that all else equal, countries
with lower median ages and also fewer older people, had reduced covid
outcomes. Please clarify this point.
3) Line 194/Figure 5: I wonder if you could jitter the dots at zero,
or in some other way help the eye see how many dots there are where
they are overlapping.
4) Line 282: the analyses for the older group may not show correlation
for other reasons, and you should note these. One is that the
variation and range of x axes is less: less variation in bcg coverage
axis means less opportunity to find the correlation. Another is that
older people would have received their BCG vaccines a longer time
prior to exposure to SARS-COV-2.
5) Line 285: this sentence is not directly supported by your data
presentation. Please flesh out your argument, since it is not clear to
this reader that the analysis supports that the BCG impacts on the
young and middle-aged populations amount to an important effect (in
terms of effect size) on the total COVID-19 burden for the
country. The evidence is perhaps all present in the article but for
this reader it needs to be more clearly step-by-step conveyed as
supporting this interpretation that the BCG effects that are
statistically significant in the relatively low-disease burden younger
population can be interpreted as meaningfully impacting the overall
disease burden of the country. The rest of this paragraph contains
multiple claims that appear to be interpretation presented as fact,
for instance line 287 ("the main contribution of the young age is with
regards to the impact on the chain of infection") - this is an
interesting interpretation but it is a causal statement not supported
by your observational analysis. Also see line 332 in which it is
stated that you can conclude from this analysis that the young
population has a "validated" role in the spread of COVID-19, which
does not appear obviously supported by your data. Likewise, Line 291 claims without citation or data support that the elderly
do not contribute to viral transmission. This does not seem to be
supported by your analysis - if it is, please flesh out the
explanation.
6) Line 288 "overlaps with the group": please communicate the age
range for this group here so that the reader won't have to look it up
in the cited source.
7) Line 312 "by altering the threshold for the alignment date" appears
mis-stated, since if I understood correctly the alignment date is
fixed across the analyses, but the subset of data are chosen by
reference to that alignment date, with varying date ranges.

Mostly Minor comments:
1) Typo on line 28 ("contribute")
2) typo line 50 ("vaccines throughout")
3) typo line 57 ("outcomes.")
4) In statistical methods section, indicate the (eg R) stat stoftware
version used, and when describing the Pearson test p-value from software, insert
the word "analytic" or even "t test" to make clear that you are
contrasting the permutation test with that usual test based on the
asymptotic approximation.
5) typo line 265-266 ("infectious diseases")
6) type line 270 ("cultural events")
7) line 271-273: sentence fragment.
8) Typo line 322: "sources of BCG" .. "efficacies".k
9) typo line 326 "monitoring the evolution"

Author Response

1) Line 124: please comment on the potential for the result to be driven by a small number of influential points. It seems from eying this and other plots that the countries with 0 BCG have higher variation and higher mean values of the covid outcomes in multiple analyses. Is there a correlation among the countries with non-zero values? If not, then perhaps stating it as a general correlation is not sufficiently clear. If so, it's useful to state this as it is not clear to the reader from the plots.

Thank you for that comment. Indeed, countries with no previous BCG vaccinations, have a higher variance in terms of BCG outcomes, as can be seen in Figure 2. Removing those countries (as you have suggested) reduced the significance of correlation dramatically. We argue that indeed much of the signal is governed from no BCG policy. However, to assess the robustness of our statistical measure, we added sub-sampling tests. The results confirmed the robustness of the correlations. 

In the revised version we added this new analysis to Materials and Methods (lines 92-94) and Results (lines 130-134 in Red). We also added a new Supplemental Table S6 for the results.

2) Line 157-158: This is a subtle point but I think I get it. Median age reflects both lifespan and demographic bulges, etc. So this negative coefficient could indicate that all else equal, countries
with lower median ages and also fewer older people, had reduced covid outcomes. Please clarify this point.

Thank you for this important comment. We added several key publications that discuss COVID-19 in view of age groups (with detailed age partitions data across major regions). We state that higher median age is a parameter reflective of lifespan and the demographic bulges and links to a higher fraction of the old population. Since the infected older population have a higher risk of death, countries with a higher median age are naturally susceptible to a higher percentage of deaths- in a matching manner with the strong positive coefficient found in the multivariable analysis.

We elaborate on the issue further in lines 153-160 of the Results in Red font.

3) Line 194/Figure 5: I wonder if you could jitter the dots at zero, or in some other way help the eye see how many dots there are where they are overlapping.

Thank you for your suggestion, we jittered the dots in Figure 5 (lines 229-231 in Red font).

4) Line 282: the analyses for the older group may not show correlation for other reasons, and you should note these. One is that the variation and range of x axes is less: less variation in bcg coverage axis means less opportunity to find the correlation. Another is that older people would have received their BCG vaccines a longer time prior to exposure to SARS-COV-2.

Thank you for the suggestion. We state that the younger age group that was recently vaccinated is likely to present immunological protection, whereas the older age group- that was vaccinated with BCG over 65 years ago- are unlikely to display vaccination driven immune protection. In addition, we added relevant citations to support this view. 

In the Results section, we also performed the age group correlation analysis while not weighting the fraction occupied by each of the age groups (not shown by a figure, but the results are fully reported). The results of such analysis affirmative the ones obtained from the weighted correlation analysis results. Of course, the x-axis, in this case, ranges from 0 to 1 and is identical for each age group. Thus, we argue that the lack of correlation for the elderly remains valid and it is not explained by the limited range of the x-axis.

We elaborated on the issue further section 3.3 of the results (lines 195-200) and in the Discussion (lines 308-318)

5) Line 285: this sentence is not directly supported by your data presentation. Please flesh out your argument, since it is not clear to this reader that the analysis supports that the BCG impacts on the
young and middle-aged populations amount to an important effect (in terms of effect size) on the total COVID-19 burden for the country. The evidence is perhaps all present in the article but for this reader it needs to be more clearly step-by-step conveyed as supporting this interpretation that the BCG effects that are statistically significant in the relatively low-disease burden younger population can be interpreted as meaningfully impacting the overall disease burden of the country. The rest of this paragraph contains multiple claims that appear to be interpretation presented as fact, for instance line 287 ("the main contribution of the young age is with regards to the impact on the chain of infection") - this is an interesting interpretation but it is a causal statement not supported by your observational analysis. Also see line 332 in which it is stated that you can conclude from this analysis that the young
population has a "validated" role in the spread of COVID-19, which does not appear obviously supported by your data. Likewise, Line 291 claims without citation or data support that the elderly do not contribute to viral transmission. This does not seem to be supported by your analysis - if it is, please flesh out the explanation.

Thank you for the suggestion and the offer to simplify the messages.. We agree that some of our statements were not sharp enough. We stress the fact that the analysis that was performed, measured the total population morbidity and mortality with regard to the age group coverage. In addition, we changed the y-axis label of Figure 5 (a-c) to “Total Population, Deaths per 1M” to clarify the results.  Lastly, to your suggestion, we added several new citations that support our claims. We also added a more elaborated discussion on the limitation of the analysis to provide a balanced and accurate view of COVID-19 global dynamics (that remains to some extent unpredicted). Of course, we cannot rule out the possibilities for other explanations for our findings (e.g. Ethnic-based susceptibility). 

We rewrote some of these sentences in the Discussion (see red font). 

6) Line 288 "overlaps with the group": please communicate the age range for this group here so that the reader won't have to look it up in the cited source.

We added the data supported by the references to the text. The study discussed children and young adults' behavior. We provided the information (line 313)

7) Line 312 "by altering the threshold for the alignment date" appears mis-stated, since if I understood correctly the alignment date is fixed across the analyses, but the subset of data are chosen by reference to that alignment date, with varying date ranges

Thank you for your note. We made a mistake and meant to refer to Figure 2- where we also conducted an analysis on different alignment dates (0.5 and  2 DPM) and showed the robustness of our analysis notwithstanding the change in the alignment date. The mistake (between the Figure numbers) was corrected in the text.

 In addition, we encourage all users to change any of the alignment dates as explained in Results section 3.4. The alignment date is a free parameter that can easily be changed.

Mostly Minor comments:
1) Typo on line 28 ("contribute")
2) typo line 50 ("vaccines throughout")
3) typo line 57 ("outcomes.")

These typos as well as others were fixed, thank you for the careful reading.

4) In statistical methods section, indicate the (eg R) stat stoftware version used, and when describing the Pearson test p-value from software, insert the word "analytic" or even "t test" to make clear that you are contrasting the permutation test with that usual test based on the asymptotic approximation.

Thank you for the note. We added the names and the appropriate citations. In addition, we added the word “analytic” 

5) typo line 265-266 ("infectious diseases")
6) type line 270 ("cultural events")
7) line 271-273: sentence fragment.
8) Typo line 322: "sources of BCG" .. "efficacies".
9) typo line 326 "monitoring the evolution" 

These typos, as well as others, were fixed, thank you for the careful reading.